# Regulatory Role of GgaR (YegW) for Glycogen Accumulation in *Escherichia coli* K-12

**DOI:** 10.3390/microorganisms12010115

**Published:** 2024-01-05

**Authors:** Shunsuke Saito, Ikki Kobayashi, Motoki Hoshina, Emi Uenaka, Atsushi Sakurai, Sousuke Imamura, Tomohiro Shimada

**Affiliations:** 1School of Agriculture, Meiji University, 1-1-1 Kawasaki-Shi, Kanagawa 214-8571, Japan; 2Research and Development Section, Diagnostics Division, YAMASA Corporation, 2-10-1 Araoicho, Choshi, Chiba 288-0056, Japan; 3Space Environment and Energy Laboratories, Nippon Telegraph and Telephone Corporation, Musashino-shi, Tokyo 180-8585, Japan

**Keywords:** transcription factor, glycogen, ADP-glucose, *Escherichia coli*, gene regulation

## Abstract

Glycogen, the stored form of glucose, accumulates upon growth arrest in the presence of an excess carbon source in *Escherichia coli* and other bacteria. Chromatin immunoprecipitation screening for the binding site of a functionally unknown GntR family transcription factor, YegW, revealed that the *yegTUV* operon was a single target of the *E. coli* genome. Although none of the genes in the *yegTUV* operon have a clear function, a previous study suggested their involvement in the production of ADP-glucose (ADPG), a glycogen precursor. Various validation through in vivo and in vitro experiments showed that YegW is a single-target transcription factor that acts as a repressor of *yegTUV*, with an intracellular concentration of consistently approximately 10 molecules, and senses ADPG as an effector. Further analysis revealed that YegW repressed glycogen accumulation in response to increased glucose concentration, which was not accompanied by changes in the growth phase. In minimal glucose medium, *yegW*-deficient *E. coli* promoted glycogen accumulation, at the expense of poor cell proliferation. We concluded that YegW is a single-target transcription factor that senses ADPG and represses glycogen accumulation in response to the amount of glucose available to the cell. We propose renaming YegW to GgaR (repressor of glycogen accumulation).

## 1. Introduction

Glycogen accumulation occurs in *Escherichia coli* and many other bacteria. Glycogen is formed when there is an excess of carbon under conditions in which growth is limited because of the lack of a growth nutrient, such as a nitrogen, sulfur, or phosphate source [1]. Glycogens play a role in prolonging the viability of microorganisms in the stationary phase by providing a source of carbon and energy [2,3]. The accumulated glycogen also plays an important role as the primary source of glucose for energy production during regrowth in the early inducible phase [4]. Recent reports have also indicated that glycogen metabolism is involved in various stress conditions, such as starvation, biofilm formation, cold stress, desiccation, and oxidative stress in *E. coli* [5].

In *E. coli*, the enzymes for glycogen synthesis and degradation are encoded by genes that constitute the *glgBXCAP* single operon: *glgC* (glucose-1-phosphate adenylyltransferase (AGPase)), *glgA* (glycogen synthase), *glgB* (glycogen branching enzyme), *glgP* (glycogen phosphorylase), and *glgX* (glycogen debranching enzyme) [6,7]. ADP-glucose (ADPG) is a glucosyl donor for bacterial glycogen synthesis, and *E. coli* glycogen synthases are specific for the sugar nucleotide ADPG [8]. Therefore, the modulation of AGPase activity is an important allosteric regulator of bacterial glycogen synthesis. The primary activator of *E. coli* AGPase is fructose 1,6-bisphosphate, and AMP is its major inhibitor [1,9,10]. Therefore, under conditions of limited growth with excess carbon in the medium, the accumulation of glycolytic intermediates, such as fructose-1,6-bisphosphate, is thought to occur and could be a signal for the activation of ADPG synthesis. Although there are many reports on the allosteric regulation of enzymes involved in glycogen metabolism, gene expression is reportedly induced during the stationary phase [7]. The rate of glycogen synthesis is inversely related to the growth rate when growth is limited by the availability of certain nutrients, such as nitrogen. Consistent with this relationship, the levels of glycogen biosynthetic enzymes in *E. coli* increase as cultures enter the stationary phase [2,3].

In transcriptional regulation, full-length *glgBXCAP* is transcribed from the *glgB* promoter, and the transcription unit of *glgCAP* is transcribed from the promoter of *glgC*, both of which are recognized by the major sigma subunit, sigma-70, encoded by *rpoD*, which plays a major role in the transcription of growth-related genes. In addition, there is another promoter upstream of *glgB* [11], which is recognized by sigma-38, encoded by *rpoS*, and plays a role in the transcription of stationary-phase-specific genes. Glycogen accumulates during the stationary phase in an RpoS-dependent manner [12]. Other transcription factors, such as the cAMP receptor protein (CRP), which senses glucose uptake activity, activate *glgCAP* but do not affect *glgBX* [11], and the PhoP-PhoQ two-component system activates *glgBXCAP* (sensing environmental Mg^2+^) [13]. At the translational level, CsrA (a carbon storage regulator) binds to *glgCAP* mRNA and inhibits its translation [14]. In addition to a set of *glg* genes, it has recently been reported that the inner membrane nucleoside transporters NupC and NupG incorporate ADPG, depending on CRP regulation [15]. Previously, the KO collection, a collection of 3985 non-essential gene-disrupted mutants of *E. coli* K-12, and the ASKA library, a gene expression library of 4123 genes, were used to screen for genes involved in glycogen metabolism [16,17,18]. This has led to the proposal of various functions related to glycogen metabolism; however, so far, the genes that directly act on glycogen metabolism are limited to the *glg* operon and its regulators, and glycogen is thought to accumulate during the stationary phase. Because glycogen production is based on glucose availability, it should be regulated not only by the growth phase, but also by the amount of glucose and similar sugars; however, such mechanisms and regulations have not yet been revealed.

*E. coli* contains approximately 4500 protein-coding sequences in its genome. *E. coli* has seven sigma factors and approximately 300 transcription factors (TFs) to precisely regulate the genes required for responses to various environmental changes [19,20]. However, even within the model bacterium *E. coli*, the function of one-fifth of this species’ genes remains unknown. Owing to the advancement of genome-wide research technologies, such as transcriptomics and proteomics, the regulatory role of each TF has been identified mainly based on transcription patterns in the absence of a test regulator or after its overexpression. A large amount of knowledge of TFs is assembled in databases such as EcoCyc [21] and RegulonDB [22], but with the use of these gene expression analyses it is difficult to distinguish between the direct regulation of test TF and their indirect influence. The regulatory targets of TFs often include genes encoding other TFs that form the hierarchy of the TF network [20]. To identify the direct regulatory targets of TFs, Chromatin Immunoprecipitataion (ChIP) [23], and Genomic SELEX analysis [24] have been widely used in vivo and in vitro, respectively. TFs bind to DNA and regulate nearby genes. Thus, predicting the regulation of target promoters, genes, and operons is possible based on the location of TF-binding sites. Using these genome-wide methods to identify the direct regulatory network of a TF and its regulated gene(s), it is possible to infer the function or role of unknown regulated gene(s) or unknown TFs based on the function of either the TF feature or the regulated gene.

To understand the overall picture of genomic transcriptional regulatory mechanisms, it is necessary to understand the functions of a set of *E. coli* TFs. As a part of our research strategy, in this study we sought to characterize YegW, which has been recognized as one of the uncharacterized TFs in *E. coli* and belongs to the GntR family. YegW has a DNA-binding domain at the N-terminus and a signal receiving domain, which is involved in interactions with the ligand at the C-terminus [19,25]. A systematic search was performed to identify the target genes controlled by YegW using the ChIP-chip method [26]. As a result, the binding of YegW was only observed at two closely located sites within the *yegTUV* upstream region on the *E. coli* K-12 genome. Although the function of *yegTUV* is unknown, its possible association with ADPG has been predicted [15], so we tested the effect of YegW on glycogen accumulation in this study. YegW is a repressor of *yegTUV*, sensing ADPG and suppressing glycogen accumulation. Therefore, we propose renaming YegW to GgaR (a repressor of glycogen accumulation).

## 2. Materials and Methods

### 2.1. E. coli Strains and Plasmids

*E. coli* BW25113 [27], its *ggaR* single-gene knockout mutant JW2088 [28], and the expression plasmid from the ASKA clone library were obtained from the *E. coli* Stock Center (National Bio-Resource Center, Chiba, Japan). Cells were grown in Luria broth (LB) medium, Kornberg medium (KM; 1.1% K_2_HPO_4_, 0.85% KH_2_PO_4_, and 0.6% yeast extract) [29] or M9 minimal medium supplemented with various concentrations of glucose at 37 °C with constant shaking at 150 rpm. When necessary, kanamycin (20 μg·mL^−1^) or chloramphenicol (30 μg·mL^−1^) was added to the medium. Cell growth was monitored by measuring turbidity at 600 nm.

### 2.2. ChIP-Chip Analysis

A ChIP assay was used to measure the chromosome-wide DNA-binding profile of GgaR using the experimental protocols described by Shimada et al. [26]. Briefly, cultures of *E. coli* BW25113 and, as a control, JW2088 Δ*ggaR* were grown to mid-log phase at 37 °C. The cells were then treated with 1% formaldehyde and broken open by sonication, which also fragmented the cross-linked nucleoproteins. Cross-linked GgaR-DNA complexes were immunoprecipitated from cleared lysates of BW25113 cells using anti-GgaR rabbit polyclonal anti-serum, and parallel samples were isolated from control JW2088 Δ*ggaR* cells. The cross-links were then reversed, and the immunoprecipitated DNA was purified. DNA samples isolated from BW25113 and control Δ*ggaR* cells were labelled with Cy5 and Cy3, respectively. To identify the DNA segments specifically associated with GgaR, the two labelled samples were combined and hybridized to an *E. coli* DNA tiling microarray [26]. For each probe, the Cy5/Cy3 ratio was measured and plotted against the corresponding position on the *E. coli* BW25113 chromosome to create a GgaR-binding profile.

### 2.3. Purification of the GgaR Protein

The plasmid (pGgaR) for the purification of GgaR was constructed as described by Shimada et al. [30]. Briefly, GgaR coding sequences were polymerase chain reaction (PCR)-amplified using *E. coli* K-12 W3110 genomic DNA as a template and inserted into the pET21a (+) vector (Novagen, Darmstadt, Germany) between the NdeI and NotI sites. The pGgaR expression plasmid was transformed into *E. coli* BL21 (DE3) cells. Transformants were grown in LB medium, and GgaR expression was induced using isopropyl β-D-thiogalactopyranoside (IPTG) in the middle of the exponential phase. The GgaR protein was purified by affinity purification using a Ni-nitrilotriacetic acid (NTA) agarose column. The affinity-purified GgaR protein was stored and frozen in the storage buffer at −80 °C until further use. Protein purity was greater than 95% as determined by sodium dodecyl sulfate-polyacrylamide gel electrophoresis (SDS-PAGE).

### 2.4. Gel Shift Assay

A gel shift assay was performed according to standard procedures [31]. Probes for the GgaR-binding target sequences were generated using PCR amplification using a pair of primers (Appendix A) and Ex Taq DNA polymerase (TaKaRa, Kyoto, Japan). A mixture of each probe and GgaR was incubated at 37 °C for 30 min in the binding buffer. When the effectors were added, the mixture was incubated for an additional 30 min. After addition of the DNA loading solution, the mixture was subjected to 5% polyacrylamide gel electrophoresis (PAGE). The DNA in the gels was stained with GelRed (Biotium, Fremont, CA, USA) and detected using LuminoGraph I (Atto, Tokyo, Japan).

### 2.5. Northern Blot Analysis

Total RNA was extracted from *E. coli* cells using ISOGEN solution (Nippon Gene, Tokyo, Japan). RNA purity was verified using electrophoresis on a 1.2% agarose gel with formaldehyde, and followed by GelRed-staining. Northern blot analysis was performed as described previously [32]. Digoxigenin (DIG)-labelled probes were prepared using PCR amplification using BW25113 genomic DNA as a template with a pair of primers (Appendix A), DIG-11-dUTP (Roche, Basel, Switzerland), dNTP, gene-specific pair primers, and Ex Taq DNA polymerase. An amount of 3 μg of total RNAs were incubated in formaldehyde-MOPS (morpholinepropanesulfonic acid) loading buffer for 5 min at 65 °C for denaturation. The samples were subjected to electrophoresis on formaldehyde-containing 1.5% agarose gel, and then transferred onto a nylon membrane (Roche). Hybridization was performed using DIG Easy Hyb system (Roche) at 50 °C overnight with a DIG-labelled probe. The membranes were treated with anti-DIG-AP Fab fragments and CDP-Star (Roche) to detect the DIG-labelled probe and the images were scanned using LuminoGraph I (Atto).

### 2.6. Observation and Measurement of Glycogen

The intracellular accumulation of glycogen was qualitatively detected using staining cell pellets with an iodine solution (0.01 M I_2_, 0.03 M KI) [29]. The same number of *E. coli* cells (OD_600_ × culture volume (mL) = 2.0) was stained with an iodine solution. Intracellular glycogen was observed using transmission electron microscopy, following the method described by Eydallin et al. [16] with some modifications. Cells were fixed with 2.5% glutaraldehyde in 0.01 M phosphate buffer (pH 7.3) for 30 min at 4 °C, rinsed in 0.1 M phosphate buffer, and post-fixed 2% OsO_4_ in phosphate buffer (pH 7.3) for 1 h at 4 °C. After three washes with pure water, the cells were resuspended several times in an aqueous solution with gradually increasing ethanol concentrations for dehydration. The cells were further resuspended several times in an ethanol solution with gradually increasing acetone concentration. The cells were then resuspended in acetone solution with gradually increasing concentrations of Quetol651 and, finally, the cells were embedded in 100% Quetol651 solution at 60 °C for two days. The ultrathin sections were cut using a diatome diamond knife. The sections were stained with 2% uranyl acetate and lead acetate, and inspected using a transmission electron microscope at 120 kV (JEM-2010).

Glycogen measurements were performed according to the method described by Iijima et al. (2021) [33] with some modifications. Equal amounts of cells (OD_600_ × cell culture (mL) = 10) were collected using centrifugation (8000× *g* at 25 °C for 5 min) and the pellets were resuspended in 100 μL of 3.5% (*w*/*v*) sulfuric acid. The cell suspensions were incubated at 100 °C for 160 min and centrifuged at 20,500× *g* at 4 °C for 1 min. Glucose levels in the supernatants were quantified using LabAssay Glucose (Wako, Osaka, Japan) and a Multiskan FC microplate reader (Thermo Scientific, Waltham, MA, USA). The total protein content of the cell suspensions was quantified using a Bradford protein assay kit (Bio-Rad) and a Multiskan FC microplate reader (Thermo Scientific).

### 2.7. Biofilm Formation Assay

Crystal violet staining was performed as described previously [34]. *E. coli* cells were grown in a KM medium with 1% (*w*/*v*) glucose at 37 °C in a 2.0 mL tube. Following incubation for 24 h, planktonic cells were discarded and the tubes were washed twice with phosphate-buffered saline (PBS) (-) and subsequently stained with 0.1% crystal violet for 20 min at room temperature. After extensive washing with H_2_O, the biofilm-bound crystal violet was extracted with 500 μL of 70% ethanol and measured at OD_595_.

### 2.8. Starvation Viability Assay

A single *E. coli* colony was picked up from KM agar plate for inoculating KM medium for overnight, which was then cultured fresh 5 mL KM medium with 1% (*w*/*v*) glucose at 37 °C with 150 rpm shaking rate until OD_600_ reached to 1.0. The inoculated culture was centrifuged at 6000× *g* at 25 °C for 3 min, and the cell pellets were washed three times with PBS buffer. The cells were then resuspended in 1 mL PBS buffer and placed in incubator at 25 °C for 0, 2, 4, 6, and 12 days, and were then spread on KM agar plates at each time point. The number of colonies formed on the plate after overnight incubation was counted and defined as viable cells compared to those on day 0. The experiments were independently repeated three times for the wild-type and Δ*ggaR* strains.

### 2.9. Real-Time (RT)-qPCR Analysis

RT-qPCR was performed according to standard procedures [35]. *E. coli* cells were inoculated into KM supplemented with glucose (0.25% or 1%) at 37 °C under aeration with constant shaking at 150 rpm. Total RNA was extracted from exponential phase *E. coli* cells (OD_600_ = 0.5) using ISOGEN solution (Nippon Gene, Tokyo, Japan). Total RNA was transcribed to cDNA using the THUNDERBIRD SYBR qPCR RT Set (TOYOBO, Osaka, Japan). Quantitative PCR (qPCR) was conducted using the THUNDERBIRD SYBR qPCR Mix (TOYOBO) and a LightCycler 96 system (Roche). The primer pairs are listed in Appendix A. The cDNA templates were serially diluted four-fold and used for qPCR analysis. The qPCR mixtures, containing 10 μL of THUNDERBIRD SYBR qPCR Mix (TOYOBO), 1 μL of each primer (5 μM stock), 7 μL water, and 1 μL cDNA, were amplified under the following thermal cycling conditions: 95 °C treatment for 2 min, 45 cycles of 10 s at 95 °C and 20 s at 55 °C, and then incubated for 20 s at 72 °C. The 16S rRNA expression level was used to normalize the varying levels of the test samples, and the relative expression levels were quantified using Relative Quantification software provided by Roche. The results are presented as the average of three independent experiments.

### 2.10. Western Blot Analysis

Western blot analysis was performed using a standard method, as described previously [36]. Briefly, *E. coli* cells grown in KM supplemented with 1% (*w*/*v*) glucose or M9 minimal medium supplemented with 0.4% (*w*/*v*) glucose were harvested in time course using centrifugation and resuspended in lysis buffer (50 mM Tris-HCl, pH 7.5, 50 mM NaCl, 5% glycerol, and 1 mM dithiothreitol), and lysozyme was added to a final concentration of 20 μg·mL^−1^. After sonication, 3 μg of cell extracts was subjected to 10% SDS-PAGE and blotted on to polyvinylidene difluoride membranes using semi-dry transfer apparatus. Membranes were first immunoreacted with rabbit anti-GgaR or anti-RpoA anti-serum, and then immunoreacted with anti-rabbit IgG and HRP conjugate. The membranes were developed using an enhanced chemiluminescence kit (Amersham Pharmacia Biotech, Buckinghamshire, UK). The intensity of the bands was analyzed using an LAS-4000 IR multicolor (Fuji Film, Tokyo, Japan).

## 3. Results

### 3.1. Search for GgaR-Binding Locations Using ChIP-Chip Screening

To reveal the regulatory role of the transcription factor GgaR (renamed YegW), we used ChIP to identify the binding of GgaR across chromosomes of growing *E. coli* cells. Thus, wild-type BW25113 and the Δ*ggaR* derivative JW2088 were grown aerobically in LB medium to an OD_600_ of 0.4. The cells were then treated with formaldehyde, and cellular DNA was extracted and sonicated to yield DNA fragments of approximately 500 bp. After immunoprecipitation with anti-GgaR antibodies, DNA fragments from BW25113 and control JW2088 Δ*ggaR* cells were purified, labelled with Cy5 and Cy3, respectively, mixed, and hybridized to the microarray. After washing and scanning, the fluorescence intensities of the Cy5 and Cy3 attached to each probe on the microarray were measured and plotted along the *E. coli* genome (Figure 1). A single peak of GgaR, binding was identified at the spacer between the *fbaB* gene and *yegTUV* operon, indicating that GgaR is a single-target TF. Downstream of *yegTUV*, *ggaR* is encoded on the reverse strand. Thus, the only target of the transcription factor, GgaR, is in close proximity to its own coding region. Detailed analysis of GgaR binding within the region corresponding to the single ChIP peak indicated the presence of two binding sites, one at a bidirectional transcription unit between the *fbaB* gene and *yegTUV* operon, and another within *yegT* open reading frame (ORF) (Figure 1).

### 3.2. Confirmation of GgaR-Binding to Its Targets

Using in vivo ChIP–chip screening, we identified two adjacent GgaR-binding sites in the *E. coli* K-12 genome (Figure 1). To experimentally confirm GgaR binding to these sites, we performed gel shift assays in vitro to detect GgaR-target DNA complexes. First, *fbaB/yegT* spacer DNA probe was tested. A gel shift assay indicated that the probe formed GgaR-DNA complexes in a GgaR concentration-dependent manner (Figure 2A).

Next, we analyzed the binding of GgaR to various segments of the *fbaB-yegT* region. We prepared six different DNA fragments and subjected these DNA probes to a gel shift assay (Figure 2B). Probes were prepared for the region from *fbaB* to *yegT* ORF, with several dozen base pairs overlapping each other (Appendix A), and their binding to GgaR was verified under the presence of different GgaR concentrations. In the presence of 5 pmol of GgaR, the formation of a complex with GgaR was observed only in probe-2, which contains 190 bp upstream of the *yegT* gene (Figure 2B). When the concentration of GgaR was increased to 10 pmol, the complex formation was also observed in probe-5, which is 571–869 bp inside the ORF of *yegT*. In contrast, probes 1, 3, 4, and 6, located upstream and downstream of each other, respectively, did not form complexes. The positions of the two probes to which GgaR was bound in vitro were consistent with the positions of the two peaks identified by the ChIP–chip screening, suggesting in vivo GgaR binding (Appendix A). The results of those in vivo and in vitro experiments were also consistent with the binding affinity with GgaR, which was higher between the *fbaB-yegT* spacer than the ORF of *yegT*. Taken together, we concluded that the target site for the initial binding of GgaR is the *fbaB*/*yegT* spacer.

### 3.3. Regulatory Role of GgaR in the Expression of the Target Genes

Based on the in vivo ChIP–chip and in vitro gel shift assay results, the involvement of GgaR in the regulation of the *fbaB* and/or *yegTUV* operon was predicted. We then carried out Northern blot analysis to examine the possible influence of GgaR on these genes determined mRNA levels in vivo for each of the predicted GgaR target genes in the presence or absence of GgaR. Wild-type *E. coli* BW25113, *ggaR*-deleted mutants, and *ggaR*-deleted mutants harboring the GgaR expression vector strains were grown in LB medium, and total RNA was extracted in both the exponential and stationary phases. Based on the gene size, the predicted *yegTUV* mRNA length should be approximately 3.3 kb. In the *ggaR*-deleted mutant, the same transcripts of *yegTUV* could were detected using *yegT*- and *yegV*-specific probes (Figure 3). The size of this transcript was estimated to be 3.3 kb as compared with the size of the 2904 nucleotide 23S rRNA and was in agreement with the size predicted for *yegTUV* mRNA. The intensity of the *yegTUV* signal was somewhat reduced by complementation with the GgaR expression vector, was not detected in the wild-type strain, and showed a similar pattern during the exponential and stationary phases. In contrast, strong signals were observed in the stationary phase for the *fbaB* gene located divergently from *yegT*. This is in agreement with a previous report that the *fbaB* transcript was noticeably reduced in the log growth phase [37], but no clear effect of *ggaR* was observed. These results indicate that the *yegTUV* operon is transcribed into a single mRNA, and this transcription is repressed by GgaR, regardless of the growth phase.

### 3.4. Effect of GgaR on Glycogen Accumulation

To date, GgaR is a functionally unknown transcription factor; however, this study shows that it is a repressor of the *yegTUV* operon. Additionally, the functions of *yegT*, *yegU*, and *yegV* were unknown. Of these genes, *yegV*, the putative sugar kinase, was included in a list reported by Almagro et al. for genes whose deletion significantly reduced glycogen accumulation in KM [15]. Based on the protein sequence, YegT is predicted to belong to the Nucleoside:H+ symporter (NHS) family within the major facilitator superfamily (MFS) [38], and YegU is predicted to be an ADP-ribosylglycohydrolase [39]; any of these predicted enzymatic reactions could act on ADPG. Considering that in *E. coli*, glycogen is produced via ADPG converted from glucose as a substrate, the predicted functions of *yegTUV* genes were implicated in glycogen accumulation.

In this study, we investigated the effect of *ggaR* deficiency on cell growth and glycogen accumulation in KM, a rich medium that allows *E. coli* to enhance the accumulation of glycogen and the typically and commonly used medium in studies of glycogen metabolism in *E. coli* [29]. First, the growth curves of *E. coli* wild-type and *ggaR*-deficient strains in KM at various glucose concentrations were observed. The maximum turbidity increased with the addition of 0.25% to 1% glucose, but the deletion of *ggaR* did not affect growth (Figure 4A). Under the same conditions, the amount of intracellular glycogen was qualitatively determined by iodine staining of the collected cell pellets (Figure 4B). No significant glycogen accumulation was observed at any glucose concentration at the beginning of cultivation or without the addition of glucose. In contrast, after 6 h of incubation, glycogen accumulation was dependent on glucose concentration. Additionally, the *ggaR*-deficient strain stained more intensely than the wild-type strain at the same glucose concentration. The amount of glycogen accumulated during cell growth in medium with low concentrations of glucose, such as 0.25% and 0.5%, decreased after 24 h, which is consistent with previous reports [5,16]. In further experiments, the amount of glycogen that accumulated in the cells was quantitatively measured and compared (Figure 4C). At 6 h of growth in KM with 1% glucose, the amount of glycogen relative to total protein was measured to be approximately twice as much in the *ggaR*-deficient strain at 15 μg/mL compared to 8 μg/mL in the wild-type strain. The effect of *ggaR* deficiency was largely suppressed by complementation of *ggaR* with the harboring of a GgaR expression vector. After 24 h, glycogen accumulation increased in all strains, with only a slight increase in the *ggaR*-deficient strain compared to other strains. TEM was performed to directly observe the accumulation of intracellular glycogen granules (Figure 4D). The wild-type strain cells only had a few glycogen granules at the edge of cytoplasm at 6 h after inoculation in KM with 1% glucose, whereas large amounts of glycogen granules were observed in the *ggaR*-deficient strain. Similar observations after 4 h of cultivation revealed almost no glycogen granules in the wild-type strain, whereas some glycogen granules were observed in the *ggaR*-deficient strain (Appendix A). Taken together, these results suggest that GgaR downregulates glycogen accumulation in *E. coli* grown in KM medium in response to the addition of glucose.

### 3.5. The Influence of GgaR under Stress Conditions

Glycogen is an energy storage compound that plays an important role in bacterial environmental viability. Wang et al. [5] reported that *E. coli* strains lacking genes involved in glycogen synthesis or degradation promote the ability to form biofilms, and that strains that accumulate glycogen have an increased survival rate after starvation. Therefore, we also examined the effects of deficiency of the transcription factor GgaR, which represses glycogen accumulation. *E. coli* wild-type and *ggaR*-deficient strains were grown in plastic tubes in a static culture in KM medium supplemented with 1% glucose. After 24 h of incubation, the amount of biofilm formed on the inner tube surface was determined and quantified using crystal violet staining. The results indicated that biofilm formation was enhanced in the *ggaR*-deficient strain compared to that in the wild-type strain, with an approximately four-fold difference (Figure 5A). The survival rate after starvation was less than 60% for the wild-type strain after two days, whereas it was more than 80% for the *ggaR*-deficient strain (Figure 5B). The survival rate decreased with time, and this difference decreased as the number of days passed. After 12 days, there was almost no difference, a pattern similar to that reported for glycogen metabolism in gene-deficient strains [5]. These results indicate that the effects of the deletion of *ggaR* were generally consistent with the previously reported deletion of *glg* genes involved in glycogen metabolism in their effects on the ability of *E. coli* to adapt to the stress environment [5].

### 3.6. Search for Effectors Controlling GgaR Activity

In *E. coli*, glycogen is synthesized from glucose in the culture medium as a substrate, the imported Glucose 6-phosphate is converted to Glucose 1-phosphate and then to ADPG, which is the substrate for glycogen elongation. Considering that the transcription factor GgaR is involved in glycogen accumulation, as shown by the experiments of this study, and the predicted function of the genes constituting the *yegTUV* operon, the single target of GgaR on the *E. coli* genome is *yegT* for the putative nucleoside symporter, *yegU* for the putative ADP-ribosylglycohydrolase, and *yegV* for the putative sugar kinase. The effector of GgaR is thought to be ADPG; thus, we tested the possible influence of ADPG on the regulation of the *yegTUV* operon. The potential influence of ADPG on GgaR binding to its target was examined in vitro using a gel-shift assay (Figure 6). After adding increasing concentrations of ADPG, the complex of GgaR and the *fbaB*/*yegT* spacer probe was reduced, resulting in an increase in the number of free DNA probes. To examine the specificity of the inducer, glucose and ADP were also tested for their potential influence on the GgaR–probe DNA interactions. None of the compounds interfered with the binding of GgaR to *fbaB*/*yegT* spacer probes.

To verify the effect of GgaR activity in vivo, the amount of GgaR target mRNA was measured using RT-qPCR. First, to confirm the validity of the experimental method, the target mRNA levels in the wild-type and *ggaR*-deficient strains grown in KM with 1.0% glucose were compared (Figure 7A). An approximately 30-fold increase in *yegTUV* mRNA levels was observed in the *ggaR*-deficient strain compared to that in the wild-type strain. Notably, *fbaB* mRNA, which is divergently located from the *yegTUV* operon, was not affected by the deletion of *ggaR*. These results are in agreement with the effects of GgaR on the single-target *yegTUV* in LB medium, as shown in Figure 3, suggesting that GgaR represses the *yegTUV* operon, similar to the effect observed in the KM medium. Next, we tested the effect of ADPG; however, it was difficult to obtain an ample sample of ADPG to add to the liquid medium. Therefore, we tested the effect of GgaR regulation on the concentration of glucose, the precursor of ADPG, added to KM and inoculated the wild-type strain. The results showed that the level of *yegTUV* mRNA increased approximately three-fold when 1% glucose was added to the medium, compared to when 0.25% glucose was added (Figure 7B). However, compared to 0.25% and 1% glucose concentrations, the *yegTUV* mRNA levels in the *ggaR*-deficient strain, in which GgaR was absent, did not change. These results suggest that the increase in the mRNA level of *yegTUV* upon the addition of higher concentrations of glucose to the medium in the wild-type strain was GgaR-dependent, implying that GgaR becomes inactive in the presence of ADPG and derepresses the *yegTUV* operon.

### 3.7. Physiological Role of GgaR on E. coli Growth and Glycogen Accumulation in Minimal Medium

In this study, the effects of GgaR on glycogen accumulation were tested using KM, a rich medium that promotes glycogen accumulation in *E. coli* and is typically used in research on glycogen metabolism. Deletion of *ggaR* results in an inability to repress the *yegTUV* operon and promote glycogen accumulation, but no effect on growth was observed in KM despite the storage of glucose uptake in the *ggaR*-deficient strain. Therefore, to further clarify the physiological role of GgaR, we verified the importance of the GgaR-mediated regulation of glucose consumption and accumulation as glycogen using minimal glucose medium. *E. coli* wild-type and *ggaR*-deficient strains were cultured in M9 minimal medium with different concentrations (0.2%, 0.4%, and 0.8%) of glucose, and their growth curves were obtained (Figure 8A). The results showed that both strains grew well in the presence of 0.4% glucose, a concentration commonly used in *E. coli* cultures, and grew slightly poorly in the presence of 0.8% glucose. As expected, a marked decrease in growth was observed in the *ggaR*-deficient strain at all glucose concentrations compared with the wild-type strain in minimal medium. Next, glycogen accumulation was measured in each strain during the growth phase under these conditions. The results showed that glycogen accumulation was approximately three-fold higher in the *ggaR*-deficient strain at all glucose concentrations (>80 μg/mg) compared to approximately 30 μg/mg in the wild-type strain (Figure 8B). These results indicated that in the *ggaR*-deficient strain, glycogen accumulation was increased by glucose utilization in exchange for impaired cell growth under growth conditions in which the carbon source was dependent on glucose. These results indicate that GgaR plays an important physiological role in the utilization of glucose, determining whether it is consumed for energy production or stored as glycogen.

### 3.8. Protein Expression Level of GgaR Is Constant in Growth Phase

GgaR is a single-target TF that senses ADPG and represses the *yegTUV* operon, regardless of the growth phase. Therefore, to validate the intracellular expression of GgaR at the protein level, we measured its expression over time using antibodies against rich KM and poor M9 minimal media. The control protein was the α subunit of RNA polymerase, whose expression per cell has been reported to be 5000 molecules with virtually no fluctuation throughout cell growth [40,41]. As a result, the protein level of GgaR remained roughly constant in both KM and M9 minimal media with 1% glucose during the 24 h incubation of *E. coli* wild-type strains (Figure 9A). The negative control, the JW2088 *ggaR*-deficient strain, did not express GgaR. The expression level of GgaR per cell was calculated by comparing it to the expression level of RpoA and was estimated to be approximately 20–30 molecules per cell, regardless of the medium (Figure 9B).

## 4. Discussion

Understanding the regulatory networks of genome transcription involving all seven sigma factors and all 300 TFs and identifying the association between each of these regulatory proteins and their direct targets is a major issue. We have previously elucidated the functions of more than 10 TFs of unknown function based on the identification of genomic transcriptional regulatory networks [42,43,44]. Here, we identified that the hitherto uncharacterized *E. coli* TF YegW (referred to here as GgaR) regulates a single target of the *yegTUV* operon in the *E. coli* genome and was therefore identified as a single-target regulator [42,45]. Most single-target TF genes in the *E. coli* genome are located in close proximity or adjacent to their regulatory target genes, forming a gene organization in which TFs and their regulatory targets exist as an adjacent set [45]. For GgaR, the *ggaR* gene is a typical single-target regulator located adjacent to the single-target *yegTUV* operon (Figure 1). Because single-target regulators have only a single target in the genome, their intracellular concentrations are considerably lower; for example, the expression of LacI, a single-target regulator known to regulate the lactose operon [42], has been reported to be less than 10 molecules in a cell [46]. Most transcription factors of the GntR family, to which GgaR belongs, form homodimers [25]. The intracellular concentration of GgaR with two binding sites in close proximity to the *yegT* promoter was approximately 20 molecules at constant concentration and approximately 10 molecules as a homodimer (Figure 2 and Figure 9). This low intracellular concentration of GgaR is consistent with typical features of single-target regulators.

Although no clear function is known for any of the genes of the *yegTUV* operon, the single target of GgaR, the loss of *yegV* was screened to reduce glycogen accumulation [15], and the inferred function of any of the genes appeared to be related to ADPG, such as *yegT*, the putative transporter of nucleoside, *yegU*, the putative ADP-ribosylglycohydrolase, and *yegV*, the putative sugar kinase. In accordance with predictions based on this somewhat ambiguous information, we tested the effect of GgaR on glycogen accumulation in the KM, which has been typically used in previous studies on glycogen accumulation. As expected, the loss of *ggaR* increased glycogen accumulation (Figure 4), suggesting that ADPG inactivated GgaR as an effector (Figure 6 and Figure 7). Although the expression of GgaR was constant in *E. coli* grown in KM (Figure 9), the effect of GgaR on glycogen accumulation was greater in the exponential phase and smaller in the stationary phase (Figure 4B,C). This may be because, as has long been thought, glycogen is largely produced by growth arrest or an excess of carbon sources due to the depletion of certain nutrients, such as nitrogen, and glycogen accumulates during the stationary phase by GlgBXCAP [1,3]. In the stationary phase in nutrient-rich medium conditions, GgaR played only a partial role. The regulatory role of GgaR is not specific to the growth phase, such as the growth arrest or stationary phase [1,2,3], but is dependent on the amount of intracellular carbon source by sensing ADPG. This is supported by the fact that under culture conditions in M9 minimal medium with glucose and dependence on glucose as the carbon source, the deletion of *ggaR* resulted in the accumulation of glycogen in exchange for poor growth (Figure 8). This suggests that the large amount of yeast extract added to the KM medium provided *E. coli* with sufficient nutrients for growth, such that even though GgaR was not present and glucose was stored inappropriately, glycogen accumulation did not affect the growth of *ggaR* mutant (Figure 4A). This indicates that GgaR plays an important physiological role in conditions in which the destination of glucose is important, such as whether it is consumed for energy production, necessary for cell proliferation, or stored as glycogen (Figure 10). Taken together, we propose that GgaR is a transcription factor that senses ADPG and represses glycogen accumulation in response to the amount of glucose available to the cell, and should thus be renamed as a repressor of glycogen accumulation.

Although screening for the genes involved in the glycogen metabolism has been conducted several times for all genes in *E. coli* [15,16,17,18], *ggaR* has not been identified. This was probably because they were all determined from steady-state glycogen accumulation in KM supplemented with glucose. This may be because they were all based on steady-state glycogen accumulation in KM, a nutrient-rich medium, and not under conditions requiring strict glucose utilization. The conservation of *ggaR* is high among Enterobacterales such as *Shigella*, *Citrobacter*, *Salmonella*, and *Vibrio*, suggesting that it plays an important role in adaptation to the gut environment. Recently, Gayán et al. [47] reported that the lack of *ggaR* caused *E. coli* to gain high hydrostatic pressure (HHP) resistance, which was RpoS-independent. Glycogen metabolism is important for the acquisition of various stress tolerances ([5]; see also Figure 5), and HHP resistance may be a part of this. Functional elucidation of the GgaR target genes, *yegT*, *yegU*, and *yegV*, will be helpful in understanding glycogen metabolism and awaits future studies.

## Figures and Tables

**Figure 1 microorganisms-12-00115-f001:**
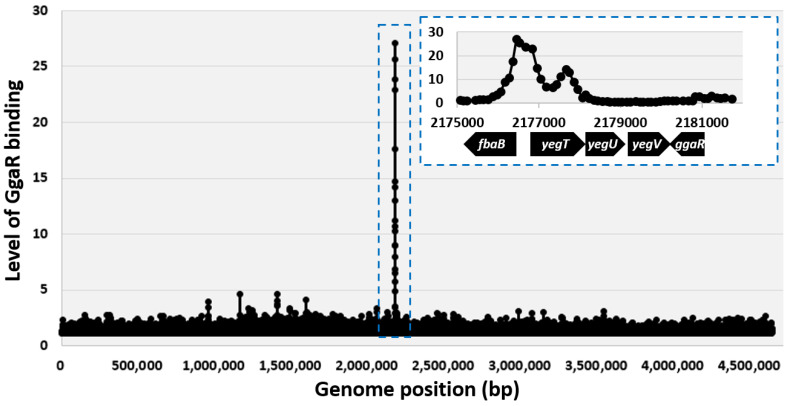
Distribution of GgaR binding across the *E. coli* K-12 chromosome. The figure shows an overview of results from the ChIP–chip experiments that measure the profile of GgaR binding across the *E. coli* K-12 chromosome during exponential growth. Binding signals (*y*-axis) are plotted against their location on the 4.64 Mbp *E. coli* chromosome (*x*-axis). A single GgaR-binding peak was identified and is indicated by the blue dotted box. The expanded region of the peak with neighboring genes is shown in the upper right blue dotted box. One peak is located within the spacer between the *fbaB* gene and the *yegTUV* operon and another is within *yegT* ORF.

**Figure 2 microorganisms-12-00115-f002:**
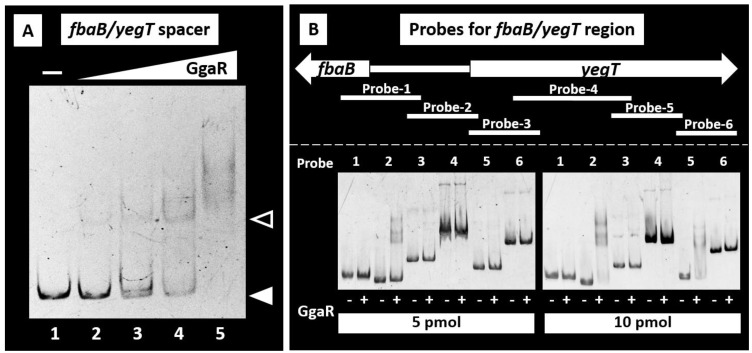
Gel shift assay of the GgaR-DNA complex formation. (**A**) Binding of GgaR to its predicted targets, the *fbaB*/*yegT* spacer, was examined. Then, 0.5 pmol of the target probe was mixed with purified GgaR. The concentration of GgaR added to lane 1, 2, 3, 4, and 5 was 0, 0.25, 0.5, 1, and 3 pmol, respectively. Filled triangles indicate the GgaR-DNA probe complex, whereas open triangles indicate free probes. (**B**) Mapping of the binding site of GgaR was carried using six different probe segments in *fbaB-yegT* region. The position of each probe in its region is shown at the top. An amount of 0.5 pmol of each probe was incubated with the absence or presence of 5 or 10 pmol of purified GgaR, and subjected to PAGE.

**Figure 3 microorganisms-12-00115-f003:**
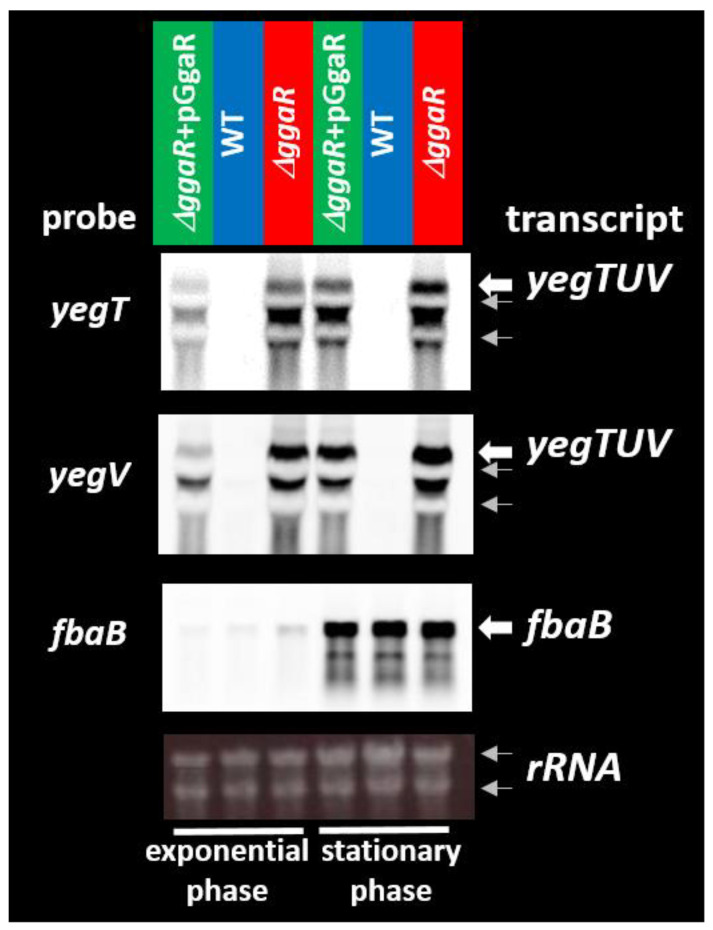
Northern blotting analysis of mRNAs from GgaR target genes. Wild-type *E. coli* K-12 BW25113, its *ggaR* mutant JW2088, and the *ggaR* mutant strain harboring the GgaR expression vector were grown in LB medium at 37 °C. Total RNA was prepared at log phase and stationary phase and subjected to Northern blotting analysis. DIG-labeled hybridization probes are shown on the left side of each panel, and the detected transcripts are shown on the right side. The positions of 23S rRNA (2904 bases) and 16S rRNA (1542 bases) were observed as a native shadow band and are indicated using gray arrows. The amounts of total RNA analyzed were examined by measuring the intensity of ribosomal RNAs.

**Figure 4 microorganisms-12-00115-f004:**
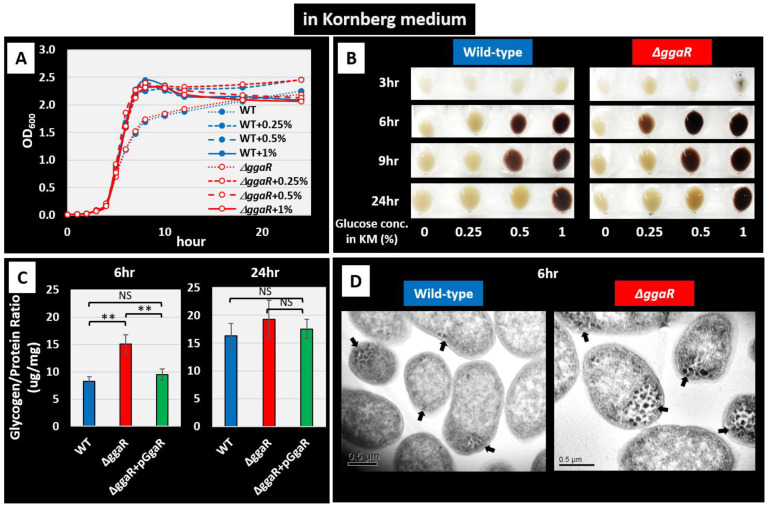
Influence of *ggaR* deficiency on glycogen accumulation in Kornberg medium. *E. coli* wild-type strain BW25113 and its *ggaR* mutant JW2088 were grown in KM with different concentrations of glucose (0, 0.25, 0.5, and 1.0%). (**A**) Cell growth was monitored by measuring the turbidity at 600 nm over time. (**B**) Iodine staining for measuring the glycogen accumulation of each wild-type strain and *ggaR* mutant cell pellets at different timepoints (3, 6, 9, 24 h). (**C**) Quantification of glycogen accumulation in *E. coli* strains (wild-type strain, *ggaR* mutant, and the *ggaR* mutant strain harboring the GgaR expression vector) were cultured in KM with 1.0% glucose and harvested at 6 h or 24 h after the start of inoculation. The glycogen amount represents the mean ± SD of three independent experiments. Statistical significance was analyzed using the Student’s *t*-test with multiple comparisons’ correction and were represented by asterisks (** *p* < 0.01). NS indicates not significant. (**D**) TEM analysis of glycogen granules. *E. coli* strains were cultured in KM with 1.0% glucose and harvested 6 h after the start of inoculation. Arrows indicate the position of glycogen granules.

**Figure 5 microorganisms-12-00115-f005:**
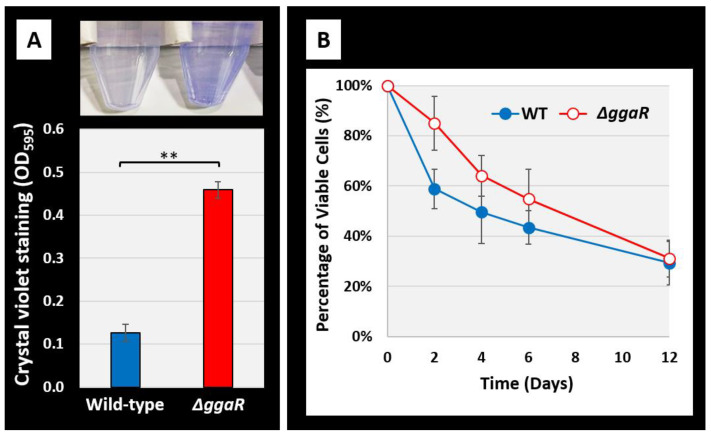
Influence of *ggaR* deficiency on survival rate under stress conditions. (**A**) Comparison of biofilm formation abilities of *E. coli* wild-type strain BW25113 and its *ggaR* mutant in KM with 1.0% glucose at 37 °C for 24 h in a plastic tube. The level of biofilm formation was measured using crystal violet staining. The upper panel shows the stained biofilm formed on the bottom of tube, and the lower panel shows the biofilm volume measured by elution with 70% ethanol. The relative level of biofilm represents the mean ± SD of three independent experiments. Statistical significance was analyzed using the Student’s *t*-test with multiple comparisons’ correction and were represented by asterisks (** *p* < 0.01). (**B**) Viabilities of *E. coli* wild-type strain BW25113 and its *ggaR* mutant under starvation conditions.

**Figure 6 microorganisms-12-00115-f006:**
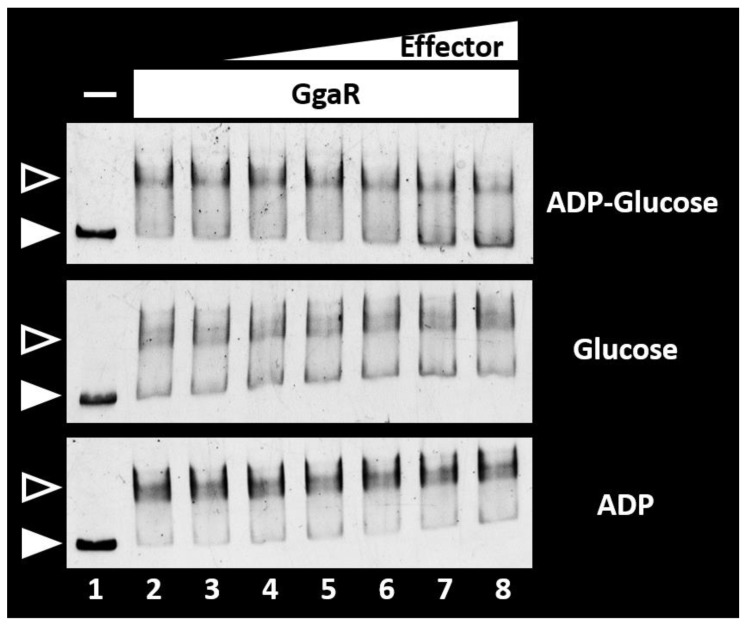
Search for GgaR ligands. The gel shift assay of GgaR-DNA complex formation and influence of each chemical. Purified GgaR was mixed with 0.5 pmol of each DNA probe corresponding to the *fbaB*/*yegT* spacer region. In the presence of 2.0 pmol GgaR (lanes 2–8), 0, 10, 20, 30, 40, 50, and 100 μM of ADPG, glucose, and ADP was added to lanes 1 and 2, lane 3, 4, 5, 6, 7, and 8, respectively. The filled triangles indicate the GgaR-DNA probe complex, whereas the open triangles indicate free probes.

**Figure 7 microorganisms-12-00115-f007:**
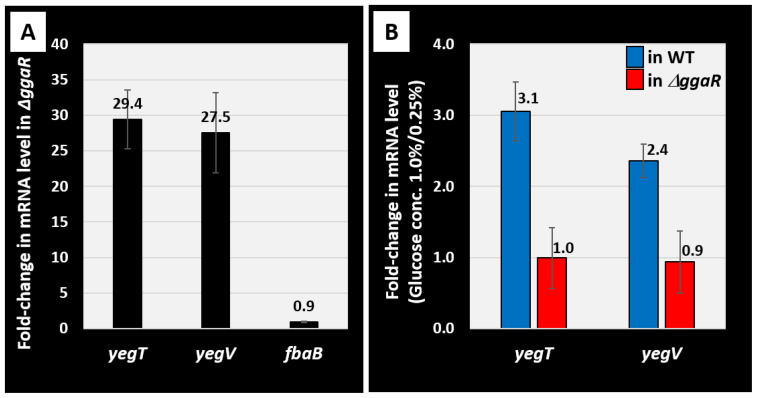
In vivo influence of *ggaR* deletion or glucose concentration in medium of GgaR targets using RT-qPCR. (**A**) *E. coli* BW25113 and its *ggaR*-deleted mutant were grown in KM with 1.0% glucose. (**B**) *E. coli* BW25113 wild-type strain and *ggaR*-deleted mutant were grown in KM supplemented with 0.25% or 1.0% glucose. Total RNA was extracted from exponential phase cells (OD_600_ = 0.5) and subjected to RT-qPCR analysis. The *y*-axis represents the relative level of mRNA of each GgaR target gene between the wild-type and *ggaR* mutant (**A**) as well the relative level between in KM with 0.25% glucose and with 1.0% glucose in the wild-type or *ggaR*-deleted mutant (**B**), respectively; the ratio of 16S rRNA is set as an internal control between the compared strains. Each experiment was repeated three times, and the average means are shown.

**Figure 8 microorganisms-12-00115-f008:**
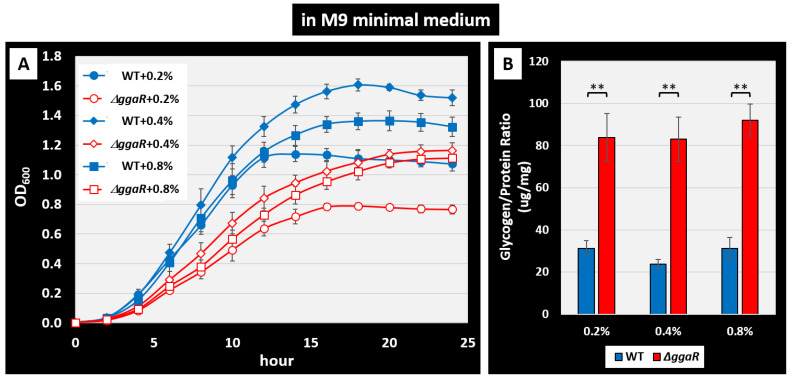
Influence of *ggaR* deficiency on cell growth and glycogen accumulation in glucose-supplemented M9 minimal medium. (**A**) *E. coli* BW25113 (blue closed symbols) and its *ggaR*-deleted mutant (red open symbols) were grown in M9 minimal medium with various concentrations of glucose: 0.25 (circle), 0.45 (diamond), and 0.85 (square). Cell density was monitored every 2 h and plotted along the time course. (**B**) Quantification of glycogen accumulation in *E. coli* wild-type and ggaR-deficient strains. The strains were cultured in M9 minimal medium supplemented with 0.25, 0.45, or 0.85 glucose and harvested at 10 h after the start of inoculation. The glycogen amount represents the mean ± SD of three independent experiments. Statistical significance was analyzed using the Student’s *t*-test with multiple comparison correction and is represented by asterisks (** *p* < 0.02).

**Figure 9 microorganisms-12-00115-f009:**
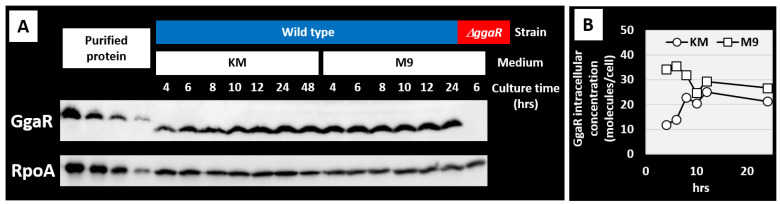
Intracellular levels of GgaR and RpoA. (**A**) *E. coli* BW25113 and its *ggaR*-deleted mutant were grown in KM with 1.05 glucose- or 0.45 glucose-supplemented M9 minimal medium and subjected to quantitative Western blotting analysis for determination of GgaR and RpoA intracellular levels. The left four lanes for the standards curve of purified proteins: His-tagged GgaR, 1, 0.5, 0.25, 0.1 ng; RpoA, 120, 60, 30, 10 ng. Each 3 μg of whole-cell extracts of cells was harvested at culture time. (**B**) The concentrations of GgaR monomer were calculated based on the concentration of RpoA monomer that stayed constant throughout the culture at the level of 5000 molecules per genome equivalent of DNA.

**Figure 10 microorganisms-12-00115-f010:**
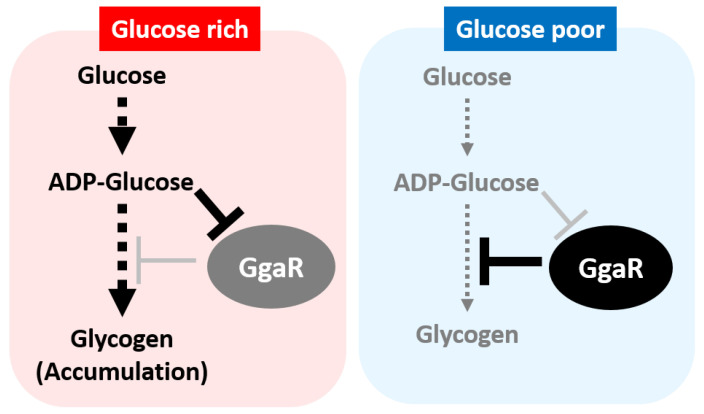
Model of regulatory network of ADPG-sensing glycogen accumulation repressor GgaR in *Escherichia coli* K-12. Under the presence of glucose, the concentration of ADPG increases, GgaR becomes inactive, glycogen accumulation is de-repressed, and glycogen accumulates. In contrast, in the absence of glucose, the concentration of ADPG decreases, GgaR becomes active, and glycogen accumulation is repressed.

## Data Availability

ChIP–chip data for GgaR have been deposited in the ‘Transcription factor profiling of *Escherichia coli*’ (TEC) database at the National Institute of Genetics (https://shigen.nig.ac.jp/ecoli/tec/ (accessed on 11 December 2023)).

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
