# Peer review of "Regulatory Role of GgaR (YegW) for Glycogen Accumulation in Escherichia coli K-12"

_microorganisms, 2024, doi:10.3390/microorganisms12010115_

Round 1

Reviewer 1 Report

Comments and Suggestions for Authors

Major comments

1. In this article, the black and white pictures are not clear enough, change the pictures to a bright color, so that the important data information can be more visually highlighted.

2. In the results, in the figure1, it should be marked the peak, and a detailed explanation.

3. In the results, the result plots of RT-qPCR are not shown.

4. In the methods, it is mentioned that the image is analyzed using LAS-4000 IR multicolor, but this image and description are not presented in the results and discussion.

Minor comments

1. Line 71, please change the spacing between words.

2. Line 82, the , should be deleted.

3. Line 97, please delete the space after "TFs".

4. Line 198, 220, 290, 395, 445, please delete a space.

5. Line 228, please list the primers used here.

6. Line 292, what are the six different probes?

7. Line 310, please revise conclude as concluded.

8. Line 332, the that should be deleted.

9. Line 346, please revise reduces as reduced

10. Line 356, a reference should be added.

11. Line 377, the , should be deleted.

12. Line 391, please revise 6 as 6h.

13. Line 395, 424, 514, please revise are as were.

14. Line 414, a reference should be added.

15. Line 541, the predicate is in the wrong form.

16. Line 574-576, a reference should be added.

17. Line 583-584, a reference should be added.

Comments on the Quality of English Language

Manuscript writing should be improved.

Reviewer 2 Report

Comments and Suggestions for Authors

Saito et al. provides a scientifically sound manuscript demonstrating the function of YegW in Escherichia coli K12.  The work is significant from several perspectives:

1. Function and regulation of the yegTUV operon has not been clearly determined.

2. The role of yevW, as described in this manuscript, shed light on the regulation (repression) of the operon

Reviewer's comments:

1. The data presented in the manuscript are comprehensive and the results are presented quite clearly. The authors methodically prove their points, convincingly.

2. The ChIP-chip experiments set the tone for the study, and I agree that it is likely that YevW is a single target transcription factor. 

3. Statistical analyses, where warranted, are convincing.

4. Figure 3: I suggest that the authors show the entire Northern blot. Were additional bands detected?

5. The authors propose renaming YegW as GgaR. The Reviewer agrees with the proposal. However, in the current manuscript, the authors should retain the designation as YegW throughout the manuscript. The proposed change should only occur in the Abstract and Discussion, and perhaps the last paragraph of the Introduction. The title should be rewritten "...YegW (GgaR)..." The change could be highlighted in subsequent papers as GgaR (formerly YegW).

Line 24: "In response to increased glucose concentration..."

Line 72: delete "collections"

Line 77: "...glucose availability..."

Line 83: "...this species genes..." or simply "genes"

Line 201: indicate the name of the specific protein assay kit used.

Line 251. "Cells" should be "Blots" ?

Line 527-528. Please elaborate on specifically why the assumption is 20-30 YegW molecule per cell relative to the level of RpoA. Are there appropriate references for the calculation, including (39)?

Overall: A thorough and insightful manuscript.

Author Response

Please see the attachment (Membrane image from Northern blotting are also attached).
